# A Domain-Adaptable Heterogeneous Information Integration Platform: Tourism and Biomedicine Domains

**Rafael Muñoz Gil [1,*] , Manuel de Buenaga Rodríguez [2], Fernando Aparicio Galisteo [1], Diego Gachet Páez [3] and Esteban García-Cuesta [4]**

1   Science, Computing and Technology Department, Universidad Europea de Madrid, Villaviciosa de Odón, 28670 Madrid, Spain; fernando.aparicio@universidadeuropea.es
2   Departamento de Ciencias de la Computación, Universidad de Alcalá, Alcalá de Henares, 28871 Madrid, Spain; manuel.buenaga@uah.es
3   Escuela Politécnica Superior, Universidad Francisco de Vitoria, 28223 Madrid, Spain; diegogabriel.gachet@ufv.es
4   Departamento de Inteligencia Artificial, Universidad Politécnica de Madrid, Boadilla del Monte, 28660 Madrid, Spain; esteban.garcia@upm.es
*   Correspondence: rafael.munoz@universidadeuropea.es

**Abstract:** In recent years, information integration systems have become very popular in mashup-type applications. Information sources are normally presented in an individual and unrelated fashion, and the development of new technologies to reduce the negative effects of information dispersion is needed. A major challenge is the integration and implementation of processing pipelines using different technologies promoting the emergence of advanced architectures capable of processing such a number of diverse sources. This paper describes a semantic domain-adaptable platform to integrate those sources and provide high-level functionalities, such as recommendations, shallow and deep natural language processing, text enrichment, and ontology standardization. Our proposed intelligent domain-adaptable platform (IDAP) has been implemented and tested in the tourism and biomedicine domains to demonstrate the adaptability, flexibility, modularity, and utility of the platform. Questionnaires, performance metrics, and A/B control groups' evaluations have shown improvements when using IDAP in learning environments.

**Keywords:** information systems; recommenders; information fusion; Semantic Web; intelligent agents

## 1. Introduction

Information integration is the method by which integral access is provided to distributed, heterogeneous information sources with different formats (structured, semi-structured, and unstructured) [1], thus affording users with the sensation of examining a centralized, homogeneous information system [2]. A major problem of access to digital information is its heterogeneity, a consequence of the notable growth it has experienced in recent years. This has given rise to a wide variety of interfaces, such as Open Database Connectivity (ODBC), web services, Structured Query Language (SQL), XML Path language (XPath), and the Simple Object Access Protocol (SOAP) [3]. Unfortunately, the solution—the development of information integration systems—is hampered by the variety of formats in which this information is stored: relational databases, eXtensible Markup Language (XML) format, proprietary applications formats, data feeds, social networks, etc. [4]. Hence, the challenge is to produce tools that can be used for the rapid development of applications that can harvest information from different sources and may be stored in different formats [5].

Information integration, understood as the study and unification of multiple sources of heterogeneous information into a single system, has a long history, [1] and, since its beginnings, the unification of database schemas has been an enormous challenge [6]. The representation of information in databases is usually structured, but, with the advent

of e-mail, webpages, and digital audio and video, a huge amount unstructured information has appeared, making it more difficult to find and access. The representation of information in XML format was an early attempt to bridge the gap between structured and unstructured information [7], and the later development of information integration technologies finally led to semantic integration, which is able to integrate databases, files in different formats, and ontologies [8]. The emergence of the Semantic Web [9] has been making an excellent revolution of smart web applications and adds reasoning capabilities for information accessing and integration. According to the World Wide Web Consortium (W3C), the Semantic Web refers to the vision of the web of linked data enabling people to create data stores on the Web, build vocabularies, and write rules. Some of the technologies used for those purposes are Resource Description Framework (RDF), SPARQL, the Web Ontology Language (OWL), or Simple Knowledge Organization System Namespace (SKOS).

To effectively use heterogeneous sources of information in large information systems, such as the Internet, advanced and automatic systems of information acquisition, organization, mediation, and maintenance are required. These automated agents are designed to make resources available in a proactive fashion, resolve the difficulties of accessing information, and offer value-added services, such as automatic recommendation [10]. Intelligent agents provide knowledge on a chosen topic based on pertinent information found in different information sources. These form the nucleus of information integration systems capable of collecting information and making appropriate recommendations [11].

One major difficulty that information integration systems face is the integration of information that may differ depending on the domain. Moreover, most of the information is provided in text expressed in natural language and sometimes using specific context symbols or expressions, as in social media. To process them, there is branch of computer science called natural language processing (NLP) that gives computers the ability to understand text as human beings do. Therefore, a domain-adaptable platform that uses NLP is an appealing solution for this context. The design and development of recommendation systems to complement the information integration systems links areas such as machine learning, data mining, and information recovery [12], and the resulting systems need to analyze the data accessing the complete dataset (being centralized) to exploit the big data capabilities. Hence, the creation of mashups in an enriched format is needed to analyze distributed and heterogenous data.

While data integration has been historically linked with expert owners of data to connect their data together in well-planned, well-structured ways, mashups are about allowing arbitrary parties to create applications by repurposing a number of existing data sources without the creators of those data having to be involved. In this context, linked data provides some relevant characteristic for the creation of the mashup and its exploitation. Linked data refers to a set of best practices for publishing and linking structured data on the Web and allows the re-definition of concepts to make data format homogeneous and interconnects resources facilitating the access to a set of databases using the resource description framework [13]. As far as the authors know, there has not been any work that gives a holistic perspective of the current technologies and applications of linked data mashups as well as the challenges for building linked data mashups [14].

The overall goal of our work is to achieve the integration of both structured and unstructured information sources by constituting an architecture that can be adapted to different domains and that could accommodate different technologies of the area of machine learning. We propose a platform architecture that is information domain-adaptable, similar to those described by other authors [15], including not only the specific functionalities but also the technology that makes them a complete framework. Moreover, the proposed architecture overcomes some of the problems related with centrality and the use of a unique central data system by:

- Implementing a set of functionalities that deals with heterogeneous information by using NLP technologies and concept recognition.

- Meeting W3C Semantic Web criteria. Most mashups applications do not use W3C standards and cannot be automatically accessed, reducing their functionality.
- Automatically incorporating machine learning higher-level functionalities by integrating the recommendation of information and enriching this recommendation via sentiment analysis.

This architecture has been validated in two different domain use cases—(i) a tourism domain, and (ii) a biomedicine domain. Moreover, in order to validate the end-user experience and the provided functionalities, we performed some functionality and performance tests. The tourism use case aims to create a system that provides valuable and personalized tourism information in the format of context-dependent recommendations based on users' profiles via the integration of information from Freebase, DBPedia, and Expedia. Regarding the biomedicine use case, it aims to produce a platform that allows users to provide personalized medicine and learning via the categorization of clinical cases and the recommendation of similar cases, integrating information from MedlinePlus and Freebase. These are two specific domain applications, but the overall architecture can be applied and enhance other big data applications rather than recommendations; for instance, providing early warning systems to prevent and anticipate environmental impacts on health [16], using social media data for nowcasting flu activity [17], or drug discovery and repurposing using data and knowledge from already known drugs [18]. The selection of these two use cases was based on: (i) in the tourism domain, the consultations usually involve systems that provide personalized information and recommendations; (ii) in the biomedicine domain, the search for information includes many areas of interest and sources of information from different population groups (such as the general population, medical professionals, or medical researchers).

The rest of the paper is organized as follows: Section 2 discusses the related work and architectures; Section 3 introduces the proposed architecture; Sections 4 and 5 examine its use in the domains of tourism and biomedicine; Section 6 presents the assessment of the system by end-users; and, finally, Section 7 provides the conclusions obtained in this work.

## 2. Related Work

The concept of information integration has been studied in different ways at least during the last two decades. In this section, different types of architectures are introduced, explaining the information integration from a semantic and non-semantic viewpoint. In addition, the concept of a mashup is defined, and the most important methods and algorithms used in automatic recommendation are discussed.

Many information integration architectures are based on the well-known service-oriented architecture (SOA). This type of architectures is characterized by its flexibility with respect to resource integration and service implementation as its main advantages but provides a basic integration between the parts. In [19], the authors proposed an integration architecture that aims at exploiting data semantics in order to provide a coherent and meaningful view of the integrated heterogeneous information sources. They use five different layers, focusing on the standardization of the data to conform an inter-exchangeable format and homogenize the accessing methods. The combination of web services and software agents provides a promising computing paradigm for efficient service selection and information integration and has been deeply studied, showing is gracefulness [20]. Equally, in [21], the authors proposed an intelligent multi-agent information system (MAIS) architecture for proactive aids to Internet and mobile users. They also employed Semantic Web technologies for the effective organization of information resources and service processes in the tourist domain. AgenRAIDER [22] is another example and is designed to develop a comprehensive architecture for an intelligent information retrieval system with distributed heterogeneous data sources. The system is designed to support the intelligent retrieval and integration of information with the Internet. The current systems of this

nature focus only on specific aspects of the distributed heterogeneous problem, such as database queries or information filtering.

A more general framework can be established by integrating information systems using web-oriented integration architecture and RESTful Web Services [23], composing a method for the semantic integration of information systems. This approach follows the principles of Web 2.0, which strongly promote the simplest, most open, and best scaling software development approaches. For this purpose, it employs the web-oriented integration architecture (WOIA) that extends the concept of web-oriented architecture (WOA) that, in recent years, has been gaining attention as an alternative to traditional SOA.

Other alternative approaches to those presented above include the crowdsourcing concept to create an architecture focusing on the data source acquisition process based on the concept of the 'crowd as a sensor' and providing natural language processing, semantic, and some machine learning capabilities [16].

*2.1. Information Integration*

Information integration appears as result of the need to unite heterogeneous data, i.e., unifying structured, semi-structured, and unstructured information from different sources. This term has been discussed vastly in the literature and rests on three principles [24]:

- Data exchange. This involves the transformation of information depending on how well the database schema from which the data are extracted is defined, and on how well the destination database is defined (how the data are to be arranged).
- Data integration. The data to extract may be in databases or other sources (with other schemas), but it must all end up in a single schema.
- Peer to peer integration. All of the peers are autonomous and independent; therefore, there is not any schema.

Moreover, information integration can imply or not imply semantic integration. Over the last ten years, information integration has largely been undertaken without semantic tools, with service-orientated architecture (SOA), XML, web services, universal discovery, description and integration (UDDI), and scrapping techniques as the main tools for the development of integration platforms. Despite the fact that the appearance of new tools for handling and visualizing data has improved the integration of heterogeneous schemas, they still fail to map data correctly, leaving them in dispersed database columns or in columns where they should not be.

SOA provides links to resources upon demand, thus allowing access to resources in a more flexible manner than traditional architectures [25]. Within SOAs, web services are described in Web Services Description Language (WSDL) to define their functionality and how to invoke them and publish this information in UDDI registry format. Developers access these descriptions in WSDL and select those that satisfy their integration needs, invoking the service required via XML/SOAP messages.

Web services have proven to be an ideal tool for integrating information, largely via the standards XML, SOAP, WSDL, and UDDI [26]. In addition, the high data exchange speeds now available with mobile phones provide a new method—the mobile web service—for integrating data [27]. Although XML, SOAP, WSDL, and UDDI provide acceptable interoperability and integrity, much effort is still required to integrate information in real systems [28].

The use of semantic technology in information integration, for example the use of ontologies (information models that specify the semantics of the concepts used by defined and unambiguous heterogeneous data), increases the interoperability between sources. A good way to achieve efficient information integration and interoperability is the use of W3C standards (http://www.w3.org/standards/semanticweb/, accessed on 8 August 2021). Many integrated applications use domain ontologies as their main operational tool, accomplishing the mapping in agreement with the concepts of the ontology [29]. This structure has been used to develop systems that combine web services with an ontology

and thus facilitate access to different data sources, such as OpenStreetMap, Yahoo Where on Earth, and Wikipedia [30,31].

To remedy the data integration issues of the traditional web mashups, the Semantic Web technology uses linked data based on the RDF data model for combining, aggregating, and transforming data from heterogeneous data resources to build linked data mashups [14]. In recent years, ontologies have been defined using different web semantic languages, including RDF, resource description framework schema (RDFS), and OWL, e.g., to extract information from Wikipedia.

Moreover, a widely used source of structured information in information integration applications is Freebase [32]. However, it cannot be classified as a true ontology but rather as a kind of folksonomy since concepts are structured in schemas and expressed as types grouped by domain and properties. Many projects are currently underway that use Freebase as their main operational tool, e.g., BaseKB (http://basekb.com/, accessed on 8 August 2021), which converts Freebase into a complete ontology by converting its content into RDF. SPARQL protocol and query language is the standard for consulting RDF data repositories, such as those required in the ReSIST (http://www.resist-noe.org/, accessed on 8 August 2021) project, or for consulting RDF triplets databases. They have been used to produce a rather weak integration of information via SPARQL–SQL and, in more intensive, ways how SPARQL queries are rewritten dynamically depending on the data store to be accessed [33]. This allows for a more flexible and more automatic integration.

Finally, Semantic Web services combine web services and the Semantic Web to increase automation in the digital environment. Semantic Web services add meaning to the definitions of web services and thus improve the integration of distributed systems. In [34], the authors provided a framework for integrating information using Semantic Web services to create a semantic description of the services automatically generated by their system.

### 2.2. Mashups

Nowadays, many of the developed information integration systems are based on mashup architectures. Mashups incorporate data from different sources into a single application, bringing together different technologies and languages in order to integrate information [35]. Mashup architectures are composed of three layers:

- A provider of content (data layer). Sources usually provide their data via an application programming interface (API) or web protocols, such as really simple syndication (RSS), representational state transfer (REST), and web services. RDF modelling is performed in this layer, the data are filtered via a SPARQL query, and the output elements are grouped under a SPARQL design and then published.
- A mashup site (processing layer). This web application offers integrated information based on different data sources. It extracts the information from the data layer, manages the applications involved, and prepares the output data for visualization via languages such as Java or via web services.
- A browser client (presentation layer). This is the mashup interface. Browsers find content and display it via HTML, Ajax, Java Script, or similar functionality toolkits.

### 2.3. Recommendation Systems

A recommendation system's main objective is to provide the most suitable item that a user will like given a domain. Most of the existing recommendation systems fit into one of the following two categories: (i) content-based recommendation, or (ii) collaborative filtering (CF) systems. The first approach addresses the recommendation problem by defining a user profile model U that represents all the information available on a user. In a basic problem setup, U includes the user's preferences for a set of items, later used to describe the user's likes and dislikes. The second approach—simply abbreviated as CF—has achieved the most successful results and focuses on users' behaviors as proposed by [12] rather than on the users' characteristics. This method uses the similarities among users to discover

the latent model that best describes them and retrieves predicted rankings for specific items [36]. In both cases, the users' information can be collected explicitly via questionnaires or implicitly analyzing the online web behavior. Nowadays, due to the increase in the use of Web 2.0, social networks can also be exploited as a new source of information [37,38]. Prediction methods that incorporate some information from social networks are usually more accurate than those that do not because it allows getting a better user modelling characterization or information about users' similarities [39]. The last is known as social recommendation and has been successfully applied by different authors [40,41].

An advantage of using social information in recommendation systems is the collecting of opinions. For example, sentiment analysis can enrich the recommendations made. It is important to highlight that the social information can appear in different forms, being heterogeneous, distributed, or unstructured. These types of data have increased enormously during the last decade, leading to a need for systems that group the large amounts of information generated by user communities. In addition, the language used may be ambiguous, hindering its use.

Web 3.0 integrates ontologies and intelligent agents and is defined as a flexible and autonomous program with complete functionality that co-operates with other agents to attain goals, making decisions along the way [42]. Ontologies handle data represented semantically with the aim of organizing information in such a way that it can be exchanged easily by machines or intelligent agents capable of processing it and drawing conclusions, and in recommendation systems make use of ontologies for facilitating interactions. Centralized recommendation systems are usually found on servers, which limits their efficiency, scalability, and end-user privacy [43]. The alternative involves the distribution of recommendation systems, but the complexity of this could generate huge computational loads [44]. Multi-agent systems, which are composed of several intercommunicating and collaborating intelligent agents and are very useful in distributed systems, could help overcome such problems by focusing on community preferences rather than individual end-user preferences [45]. This architecture has been used in the LOCPAT project, a personalized recommendation system that provides recommendations to a requester in an agent network [46], and in an ACI project [47].

The architecture of the platform proposed in the present work makes use of several of the above aspects and follows the work of [48] and [49] by combining semantic and non-semantic information integration and recommendation systems. The result is a mashup that not only provides information to end-users but can also recommend further information.

## 3. Intelligent Domain-Adaptable Platform (IDAP)

The proposed IDAP architecture tries to solve some of the problems presented in the previous section by allowing access to information via the identification of concept names in a source text and being capable of searching for semantic relationships and descriptive information that link them. This helps the user to gain insight into the detected concepts and discover potentially relevant information. Moreover, recommendations can be proposed based on those user preferences and the semantic association to the searches.

It is worth highlighting the emphasis of the architecture permitting the inclusion of new sources of information and how this can be presented online, enriching the experience of end-users whenever they employ different final devices. The architecture is shown in Figure 1, and it aims to facilitate the integration of the following modules that provide the general framework and described functionalities:

- End-user flow and access module
- Natural language processing and concept recognition module
- The Semantic Web module
- Recommendation intelligent agents module
- Semantic Web service
- In the next part, we describe these modules.

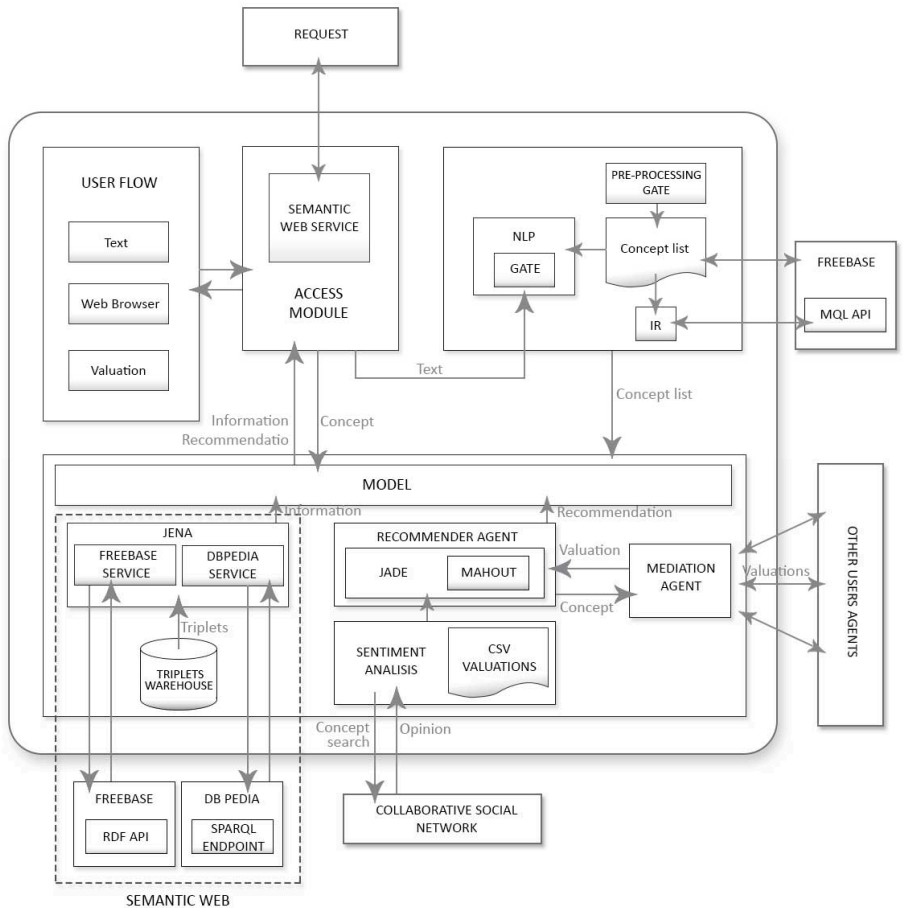

**Figure 1.** General architecture of the platform.

### 3.1. End-User Flow and Access Module

The proposed architecture has two forms of interacting with end-users or remote devices: via a web application or via a Semantic Web service (Section 3.4). Via the former, end-user flow begins by the introduction of text, and the system identifies key concepts related to the domain in which the end-user is immersed. End-users can get insights about these concepts by selecting them, obtaining information that is more detailed as well as a score (provided as a percentage) on how recommendable it is. In addition, some recommendations regarding related concepts that may be of interest are presented. The end-user can also provide feedback for those evaluations via the web application, allowing the system to learn iteratively.

### 3.2. Natural Language Processing and Concept Recognition Module

Conceptual indexing automatically organizes all the words and phrases of a text into a conceptual taxonomy that explicitly links each concept to its most specific generalizations. IDAP architecture includes the generic toolkit for text engineering, so-called GATE [50], an open-source software that provides a framework for text processing. It was conceived with the aim of facilitating scientists and developers' daily tasks by implementing an architecture for the creation of linguistic engineering applications and to provide a graphic environment in which to develop the different components necessary in these types of applications. In GATE, all the elements making up the natural language processing (NLP) software system can be classified into three types of resources:

- Language resources (LRs), which represent entities such as documents, corpora, or ontologies.

- Processing resources (PRs), which represent entities that are mainly algorithms, such as analyzers, generators, and so on.
- Visual resources (VRs), which represent the viewing and editing of graphic interface components.
- GATE provides two operational mechanisms: one graphic and one consisting of a JAVA interface. The development environment can be used to display the data structures produced and consumed in processing, as well as to debug and obtain performance measures. Among the different programming options for integrating this software into the proposed platform, we selected the development and testing with the graphical user interface (GUI) 'Developer', which makes use of the logic for the NLP module. GATE is distributed along with an information extraction system, a nearly new information extraction system (ANNIE), that incorporates a wide range of resources that carry out language analysis tasks. Gazetteer is also one of its components. Based on predefined lists, Gazetteer allows the recognition of previously mentioned concepts. These lists, in turn, allow the inclusion of features for each concept and in the present proposal are primarily used to store the Freebase identifier.

### 3.3. Use of Freebase in Semantic Access

Freebase [51] is a large collaborative knowledge database. It structures content by "topic", typically based on a Wikipedia article. A topic can be related to different things, introducing the concept of "types", which, in turn, have sets of "properties". The types are merged into "domains" (such as medicine, travel, biology, location, astronomy, music, sports, etc.), each of which is assigned an identifier (i.e., /medicine, /travel, /biology, /astronomy, /music, or /sports). Types and properties also have unique identifiers based on the concatenation of the type or topic name. For example, (A) the type gene belongs to the biology domain; hence, its identifier is /biology/gene (B), while the type tourist attraction belongs to the travel domain; thus, its identifier is /travel/tourist attraction. Freebase has already been used in Web 2.0 and 3.0 software development for name disambiguation, query classifications, and faceted browsing, among other applications.

Freebase also offers different APIs for consulting information. Among them, the search API and an API that returns data in RDF format are used to recover semantic information from each concept in Turtle format (http://www.w3.org/TR/turtle/, accessed on 8 August 2021). Freebase service handles queries are made to, and answers are supplied by, the Freebase system. The input is a chain of text that includes the name of a concept. This chain is used to perform a search of Freebase to identify the resource that corresponds to it. Once located, the RDF API is used to recover the information associated with the concept.

The different information sources used in the proposed platform provide a sufficient number of concepts for the two studied use cases, as shown in the evaluation section (Freebase itself provided 14,000 concepts for the biomedicine domain, and over 4000 for the tourism domain).

### 3.4. The Semantic Web Module

There are two important components related to semantics in the platform related to how the data is stored and how we can make inferences and reasoning with that data.

Triplet storage. All the Semantic Web-based applications possess a storage system in which RDF triplets are stored in Turtle format. This storage system is a file that initializes when the system comes online using the data in the application and content ontologies. It fills with new triplets (if there are none already in the repository) provided by the data obtained from external sources at each query.

The reasoner. This element, provided by Jena (a Java framework for Semantic Web), is used for making applications based on web semantics and linked data. The system used in the proposed platform makes use of the possibilities of Jena for:

- controlling the storage and recovery of the information (cities, attractions, means of transport, users, valuations) stored within the triplet storage system.

- handling the information recovered from Freebase to filter the data.
- converting the data recovered from DBPedia as triplets.

Jena allows the programmer to express all the queries using terms in the application ontology (a paradigm of the Semantic Web), even when the data are harvested from external sources using the corresponding content ontology. Thus, the planner supports the semantic integration of data from external sources. When SPARQL queries are made to the RDF model, the results are provided via the production of a JavaScript object notation (JSON) file. This is returned to the browser for viewing. This system is open and understandable by other systems as required by the W3C Consortium standard.

### 3.5. Recommendation Intelligent Agent's Module

All the users of the proposed platform are in possession of their own user agent. This contacts a mediating agent, the job of which is to place the user's agent in contact with those of other users. This allows the receipt of valuations from these other users and thus the formation of recommendations. As used by [52], the proposed platform employs Java agent development framework (JADE) technology to produce the user and mediating agents. Communication between these agents was achieved via Foundation for Intelligent Physical Agents (FIPA) ACL standard messaging. The automatic recommendations produced indicate valuations for different concepts (see below) and make suggestions based on (i) the relationship of these concepts to other users with similar tastes (user-based, see below), and (ii) items.

There are different criteria for measuring how similar two end-users are to each other (e.g., cosine similarity, L1, or L2). In this platform, a collaborative filtering approach has been adopted, identifying similar end-users given a user query and providing results that include item-based recommendations based on comparisons of preferences expressed by users with respect to a particular item (or concept). The recommendation engine that has been integrated in the platform is the open-source framework Apache Mahout. This software and its associated frameworks are implemented on Java and can, therefore, be executed on any modern Java virtual machine (at least Java version 6 is required), also providing easiness scalability. The data source model included in the platform is the CSV (fileDataModel).

To provide recommendations, the following parameters are defined:

- A data model: defines the input users' data, and each line has following format: userID, ItemID, Preference_value.
- A preference value: can be any real value; high values mean strong end-user preference. In the proposed model, a range between 1.0 and 10.0 was used; 1.0 indicates little interest and 10 indicates items stored as favorites.
- A similarity criterion: it measures the similarity between two different items and is defined by the Pearson correlation.
- A recommender: includes the collaborative filtering recommendation model that can be defined as item–item- or user–user-based.

### 3.6. Semantic Web Service

A Semantic Web service is a web service enriched in meta-information for facilitating searches. The incorporation of semantics to web services facilitates the automation of discovery, invocation, interoperability, and service execution. Semantic information is integrated via the annotation of web services, which can be done using a series of technologies and frameworks, the most important of which are semantic annotation for WSDL (SAWSDL) and the WOL service (WOL-S). In the proposed model, the SAWSDL service was used. This involves a small set of extensions to WSDL XML schemas that allow easier mapping of XML-based web services to a semantic model (e.g., RDF). In fact, SAWDSL defines three XML attributes that are additions to WSDL: (i) modelReference, (ii) loweringSchemaMapping, and (iii) liftingSchemaMapping. The modelReference is used to annotate a WSDL interface with entities of a semantic data model, such as URLs

for classes of an OWL ontology. LiftingSchemaMapping and loweringSchemaMapping are used to provide correspondence between types of XML data and semantic data models (and vice versa). The values of liftingSchemaMapping and loweringSchemaMapping are uniform resource identifiers (URIs) that identify documents that define the transformation of data. However, SAWDSL can work with a choice of languages when making correspondences. For example, when transformations are made between XML and RDF, the mapping necessary for "lifting transformation" can be performed using extensive stylesheet language transformation (XSLT), and SPARQL followed by XSLT used for "lowering transformations".

Summarizing, one of the characteristics of the architecture is its modularity making it flexible to change, guaranteeing that new functions can be incorporated inexpensively into the platform. This allows it to be adapted to different domains. To illustrate the adaptability of the architecture employed, the platform was prepared for the domains of tourism and biomedicine, offering new functionalities (downloads of the platforms are available at https://sourceforge.net/projects/touristmedicalface/, accessed on 8 August 2021), as shown in the next sections. The assessment of both platforms is presented in Section 6.

## 4. Tourism Domain Use Case

The enormous increase in information carried by the World Wide Web has made it particularly useful to travellers, who search for information associated with tourism destinations, attractions, such as museums and monuments, hotels, traveling, and restaurants. However, the list offered by current search engines and travel websites is usually too large, requiring the users to analyze multiple options and invest too much time to find the optimum choice [53]. Information systems that provide filtered information are, therefore, very helpful in this context. The proposed platform is able to analyze texts, filter information, and make practical recommendations (demo in http://youtu.be/AyPhzxbAoMc, accessed on 8 August 2021), improving users' experience and usability.

The tourism domain has many characteristics that make it particularly interesting for information integration and recommendation systems. The tourism lexicon uses a broad terminology taken from different areas (such as geography, economics, or art history), but there are also words related with specific areas, such as hotels and restaurants, transport, leisure, and entertainment. Therefore, a generic terminology includes large amounts of everyday vocabulary but also very specific words that include technical terms related to those areas. In addition, the users generate a large amount of data related to their preferences during regular use of the Web, which is very relevant to its homogenization and posterior analysis.

The IDAP platform has been adapted for this domain and shows the following results classified by products of interest:

- Tourist attractions: monuments, parks, museums, events, etc. This type of result also includes locations and events that are defined as attractions with changing locations.
- Accommodation: hotels, bed and breakfasts, backpacker hostels, or any place to stay.
- Travel destination: a location where a person can go for a holiday.

Freebase has been used to retrieve tourist concept lists from texts (e.g., a tourist attraction, accommodation, or travel destination from the travel domain) and to link them with semantically related content. DBPedia and Expedia have also been used to harvest information on means of transport and to obtain information about hotels for the chosen destinations, respectively. As an extension, there is another source that can be included using the same methodology, which is the semi-structured data of TripAdvisor that offers 60 million valuations and independent opinions of users of tourist services (TripAdvisor, Inc. Needham, MA, USA, 2012).

The implemented MLTour (Multi-Lingual Intelligent Platform for Recommendation in Tourism) is based on the one presented above (IDAP), and Figure 2 shows an example of the user interface for the entity of Barcelona (Spain).

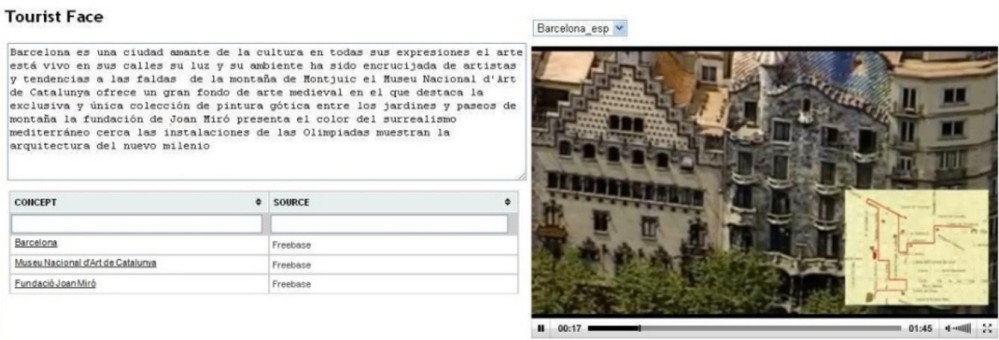

**Figure 2.** MLTour Platform Interface.

Particular emphasis has been placed on information enrichment and its online presentation to the user. The platform incorporates shallow and deep natural language processing for text analysis and components to ease the user interface generation (e.g., for mobile devices). The main purpose of this design is to render the integration of different components easier by grouping them into different modules. The modularization of the components brings together different mechanisms that allow end-users to access to the platform via HTTP protocol using the interface web or the Semantic Web service. The information retrieval module extracts the information from the runtime system, optimizing the online response time as well [54].

To adapt the IDAP platform to this domain, there is a need to define a specific application ontology that contains the terms of the following five concepts associated with the tourism domain: tourist attraction, travel destination, accommodation, hotel, and airport. For this, the Protégé (http://protege.stanford.edu/, accessed on 8 August 2021) tool has been used, an editing and ontology tool that includes a framework for the construction of intelligent systems. Freebase information source has also been used to extract information relevant to 'travel_destination', 'accommodation' and 'tourist_attraction'. These categories provide the following fields: name, description and image. The 'tourist_attraction' information was obtained via its name, and the RDF information was consulted to provide the Freebase ID of the attraction.

ML-Tour uses DBPedia to obtain transport information via its SPARQL endpoint and Apache Jena. Special emphasis was placed on the detection of airports associated with a specific destination. Elements of the type schema:Airport are sought and related to elements of the type dbpedia-owl:PopulatedPlace via the dbpprop:cityServed property. PopulatedPlace elements must have a foaf:name value that matches the name of the destination in order to obtain all the possible airports. The access to Expedia data has been achieved via the Restful API (http://developer.ean.com, accessed on 8 August 2021), and the returned XML data were transformed into RDF via XSL in order to integrate them with the data obtained from DBPedia.

The recommendations of destinations, cities, events, or locations to visit are provided by the intelligent agent recommender. It is based on the similar items collaborative filtering method (using a 1–5 ascending rating scale), and it uses the comments and evaluations of users. Table 1 shows the association of subjective labels and the obtained scores of the evaluations.

**Table 1.** Meaning of the evaluations.

| Recommendation | Score |
|---|---|
| Not recommender | From 0 to 2 |
| Little recommended | From 2 to 3 |
| Recommended | From 3 to 4 |
| Very recommended | From 5 to 4.5 |
| Impressive | From 4.5 to 5 |

The design of the Semantic Web service for the tourism domain was undertaken using Semantic Anotations for WSDL (SAWSDL). Annotations were made indicating the modelReference for the concepts City, Tourist_Attraction, Accommodation, and Transport, and the properties name, description, URL, URLimage, Transport, and webTransport.

A sentiment analysis module was added, as proposed by [55], and it analyzes comments on TripAdvisor to obtain the sentiment as either negative or positive. The module is fed using the harvested information for three categories using the Mozenda (https://www.mozenda.com/, accessed on 8 August 2021) tool: Cities, Hotels, and Attractions. From Cities, the fields City and Comments were extracted; from Hotels, the fields City, Hotel, and Comments were extracted; and from Attractions, the fields City, Attraction, and Comments were extracted. The data are converted into XML and RDF files, adhering to Semantic Web standards. The output of the module is the evaluation of the sentiment on a 1–5 ascending rating scale.

## 5. Biomedicine Domain Use Case

The biomedicine domains, and, in particular, the subdomain of personalized medicine, require flexible information systems capable of providing accurate, up-to-date, and inter-related information based on stratified access to different sources of heterogeneous data [56]. The data sources that store information relevant to the effective teaching and practice of personalized medicine can be classified into three large groups: (i) the genomic information of individual patients, (ii) the medical history (electronic health records) of a patient and others like him/her, and (iii) large biomedical databases (general and specific) able to provide information valuable in the personalization of treatment.

The resources offered by the US National Institutes of Health (http://www.nih.gov, accessed on 8 August 2021) (NIH) and by the National Library of Medicine (http://www.nlm.nih.gov/pubs/factsheets/nlm.html, accessed on 8 August 2021) (NLM) (which belongs to the NIH) include Healthfinder (http://www.healthfinder.gov, accessed on 8 August 2021) and Medline Plus, (http://www.nlm.nih.gov/medlineplus, accessed on 8 August 2021) both of which are available in English and Spanish. The resources offered to health professionals include a free search engine designed to seek out biomedical research papers (PubMed Central (http://www.ncbi.nlm.nih.gov/pmc, accessed on 8 August 2021)). The proposed platform makes use of Freebase, Medline Plus, and PubMed and is able to categorize texts using documents held at the Pathology Department of Pittsburgh University (path.upmc.edu/cases.html, accessed on 8 August 2021).

Medline has over 18 million references to articles, covering the period from 1950 to the present. It grows by 2000–4000 new references every day, and it is impossible for even the greatest of experts to keep up with all of the information being published in their fields. However, the information required for personalized medicine to be effective is not limited to that held within scientific publications; information provided by medical histories and medical reports is also required. The sources of such information are very different, and the documents they contain may differ widely in their structure. For example, they may contain free text or tables of experimental results, they may differ in length, and may be written in different languages.

The proposed platform uses the ontology Medlineplus Health Topics to access to categorize data by the identification of relevant concepts and integrates it with the knowledge on these concepts available on Freebase. Texts are processed using computational linguistic techniques, indicating whether a disease, symptom, or treatment can be referred by Medline Plus, providing additional information relevant to them. The result is a system that shortens the time needed by the end-user to understand a text of interest and increase the information about it as demanded. An earlier implementation of this module was reported and tested, obtaining successful results [57].

The implemented CLEiM (Cross Lingual Intelligent Platform for Education in Medicine) is based on the one presented above (IDAP), and it incorporates some changes to adapt it. Figure 3 shows the interface of the platform.

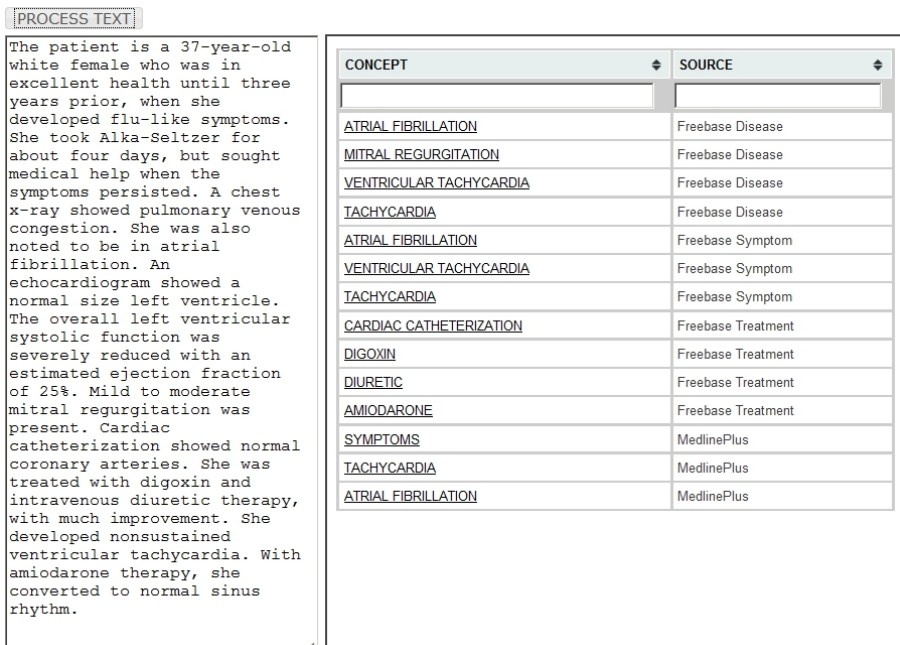

**Figure 3.** CLEiM interface.

The generic NLP, recognized concepts modules, and Freebase were used to obtain categories of diseases, symptoms, and treatments and the domain medicine to identify concepts in the examined articles. The categories used in the present demonstration were: Heart, Back, Eye, Brain, and Lung. Each category was provided with a set of frequent words, e.g., the category 'Brain' was provided with the words: concentration, planning, motor, frontal, broca, lateral sulcus, auditory area, memory, or temporal lobe.

During a search query, the categorisation process uses a similarity algorithm based on TF–IDF (term frequency–inverse data frequency) vectorization of the text to obtain the most relevant blocks of information for that query.

## 6. Assessment of the Platform and Test of Adaptability

### 6.1. Assessment of the Platform in the Tourism Domain

Data quality is desirable for any information system; good quality data are required if a system is to provide any reliable output. The problem is particularly acute for information integration systems, which harvest information from different sources with different degrees of reliability. Keeping data up to date and the maintenance of consistency and traceability are ever-harder challenges. In this context, the quality of an information system can be measured in terms of several dimensions, including originality, accuracy [58], completeness, and reliability [59,60]. Measurements can be made using formal or informal methods. Formal methods generally provide a value expressed in a unit of measurement, or a mathematical expression, while informal methods may provide a score range or a qualitative opinion assigned by a user or designer. In [61], the authors established the dimensions of 'ability to identify necessary data', 'accessibility of information', 'possibility of integration', and 'interpretation of data'. We have employed this approach for our assessment of the systems.

#### 6.1.1. Assessment of the Platform: Methodology

Based on previous work [62,63], an A/B test assessment system was designed using questionnaires that collect information on end-user satisfaction with the platform, its usability, and its effectiveness. The evaluation of the system is proposed as an activity carried out voluntarily by students in the last year of a degree in computer science. Most of them have some previous professional experience and are sufficiently knowledgeable to understand what a system of this type entails. A total of 14 students were presented to

the activity, which is considered sufficiently representative based on the low number of participants in evaluations in similar research projects, such as those seen in [63]. All the individuals received a 15-minute presentation to explain to them the platform in order to make them aware that it was an information integration system extended with a personalized recommendation system. The explanation included the information sources that the platform uses.

For the first assessment, the subjects undertook a requested 20-minute task (Task A; see Appendix A) for the tourism domain. The goal of this exercise was to obtain how long the searches took and to compare the usability of a commercial (www.visiteurope.com, accessed on 8 August 2021) and the proposed platforms.

A second assessment was performed after dividing the subjects into two groups for the A/B test. It was a 20-minute task (Task B; see Appendix A), and group A used an existing platform (YouTube), while group B used the proposed platform. Then, they were asked to fill in the questionnaire consisting of 15 questions, divided into two parts. The first part (nine questions, Q1–Q9, see Appendix B), Task A, on usability, capacity, intention, and confidence, referred to the end-user perception of the platform, while the second (six questions, Q10–Q15, see Appendix B), Task B, was about how the user felt about the functionalities related to NLP and concept recognition, information integration and sources, depth of extra information, basic recommendations, advanced suggestions, and concept evaluation entered into specific perceptions regarding the different modules making up the platform. All the questions were Likert type [64] using an ascending rating scale of 1–5 (strongly agree, agree, neutral, disagree, strongly agree).

### 6.1.2. Task A Results: End-User Usability and Capabilities Experience

Shown in Figure 4a,b are the responses obtained to questions Q1–9 (Task A) of the questionnaire. Overall, the results obtained for the end-user experience were positive. The lowest median and mode scores for the different questions were neutral, and, hence, we can affirm that most of the subjects believed that the platform helped them during the performance of the task. In contrast, the visiteurope.com platform received median and mode scores of strongly disagree for some questions (Q2, Q3, Q5, Q7, Q8).

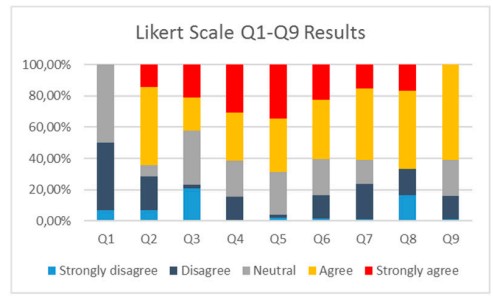

a) Questionnaire responses in Likert scale for our platform task A

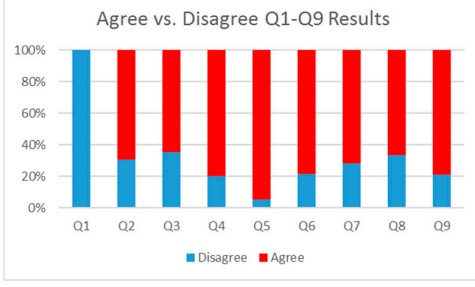

b) Questionnaire responses in agree vs. disagree scale for our platform task A

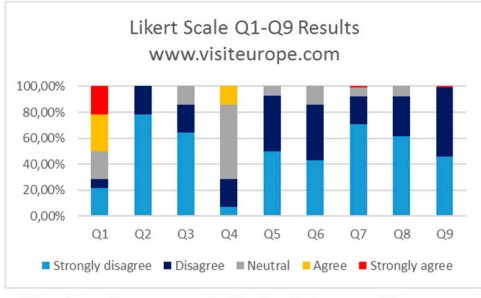

c) Questionnaire responses in Likert scale for www.visiteurope.com task A

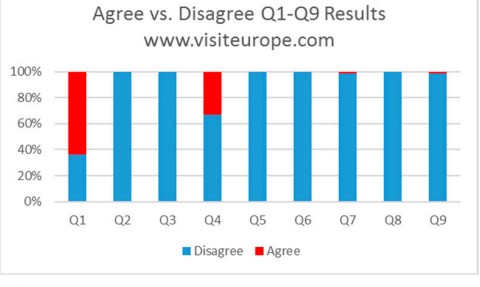

d) Questionnaire responses in agree vs. disagree scale for www.visiteurope.com task A

**Figure 4.** Results of the questionnaires in Likert and agree vs. disagree scales for IDAP (**a,b**) and www.visiteurope.com (accessed on 8 August 2021) platforms (**c,d**).

It can be observed that the users have agreed or totally agreed that the proposed platform provides optimal functionalities and capabilities for all the questions but for Q1 (is the interface friendly?), which may be due to the visual design rather than the human–machine interaction process. That is, none of the subjects agreed or totally agreed that the interface was friendly (question 1); 50% agreed, and 14% totally agreed, that the system made information easy to find (question 2); 21% agreed, and 21% totally agreed, that they were comfortable using the platform (question 3); 29% agreed, and 29% totally agreed, that the platform velocity was reasonable (question 4); and 36% agreed, and 21% totally agreed, that it was easy to learn how to use the platform. With respect to the capacity of the platform, 36% agreed, and 21% totally agreed, that the platform helped them find information (question 6); and 43% agreed, and 14% totally agreed, that the platform was faster than a traditional search system (question 7). With respect to intention, 43% agreed, and 14 totally agreed, that they would use the platform again. For comparison purposes, shown in Figure 4c,d are the results obtained for the same Task A when individuals used the website www.visiteurope.com (accessed on 8 August 2021).

6.1.3. Task A Results: Platform Functionalities

Table 2 shows the obtained results for the functionality of the platform (Q10–Q15 of Appendix B). This evaluation focuses on the following functionalities:

- Natural language processing and concept recognition (Q10).
- Information integration sources (Q11). The system recovers information from different sources (Freebase, DBPedia, Expedia, and Trip Advisor).
- Depth of information (Q12). The system shows detailed information on different concepts.
- Basic recommendation (Q13). The system indicates whether a concept is recommendable or not.
- Advanced suggestions (Q14). Depending on user preferences, the system shows new recommendations.
- Concept evaluation (Q15). The end-user can provide feedback to the system and to his/her own profile through concept evaluation.

**Table 2.** Results of the questionnaire for Task B.

| | The Functionality Is Easy to Use (a) | The Functionality Is Necessary (b) | The Functionality Is Agreeable (c) | I Was Informed about This Functionality (d) |
|---|---|---|---|---|
| Natural language processing and Concept recognition (Q10) | 57% | 43% | 64% | 57% |
| Information integration; Sources (Q11) | 79% | 64% | 50% | 50% |
| Depth of extra information (Q12) | 50% | 57% | 57% | 71% |
| Basic recommendation (Q13) | 79% | 57% | 71% | 50% |
| Advanced suggestions (Q14) | 79% | 64% | 43% | 64% |
| Concept evaluation (Q15) | 86% | 79% | 71% | 71% |

The values included in the table represent the sum of the percentages of individuals who agreed or strongly agreed with each of the functionalities. We can conclude that the end-users were satisfied with the functionality of the platform.

### 6.1.4. Task B Results: End-User Enrichment Functionalities Experience

The results obtained showed that Group A individuals made a conceptual map using an average of 17.1 concepts obtained from YouTube, while the Group B subjects made a conceptual map using an average of 15.5 concepts obtained using the proposed platform. Only 16.7% of the Group A subjects completed the task in the allotted 20 minutes using YouTube, whereas 37.5% completed the task using the proposed platform—nearly twice—and using a similar number of concepts in both cases. When they were asked about their resulting motivation to visit the destination in question or to search for further information on it, the Group A subjects returned a median and mode score of 2 (disagree) for both questions, while the Group B subjects returned a median and mode score of 4 (agree) for both questions.

### 6.2. Assessment of the Platform in the Biomedical Domain

For this domain, an evaluation has been conducted in order to gather degree teachers' opinions about the benefits of using complementary methods supported by intelligent systems. These active learning methods provide some benefits by processing texts written in natural language that help teachers to find use cases and explore similar activities in their daily teaching experience.

The study uses a mixed methodology, making use of closed questions and open ones to enable quantitative and qualitative analysis. The questionnaire is designed into five categories: (i) general information and knowledge of active teaching methods, (ii) knowledge related to concept annotation, (iii) knowledge related to cross-lingual concept extraction, (iv) knowledge related to concept/mental maps, and (v) arrangement of the evaluated IIA systems in decreasing order by perceived usefulness with a total of 66 questions [65].

The study gathered information to elucidate how useful the system is. Moreover, the proposed system, CLEiM, was compared with others of the biomedical area: BioAnnote (http://sing.ei.uvigo.es/bioannote/demo.html, accessed on 8 August 2021) and MedCMap [66]. We selected 11 teachers with different training and experience backgrounds, all of them related with Faculty of Bio-Science (BS) and Health Science (HS) at Universidad Europea de Madrid (UEM). Of the 11 BS and HS teachers interviewed, nine were women and two were men. All of them taught degree classes (seven in the Faculty of BS, two in the Faculty of HS, and two in both). Six taught medicine, five taught dentistry, four taught physiotherapy, one each taught pharmacy, optometry, and nursing. The mean age of these respondents was 42.55 years (SD = 7.34; range 30–52 years). The mean number of years of teaching experience was 10.36 years (SD = 7.50; range 2–24 years) [65]. The use of active learning methods was not common, and only one individual used sophisticated software (for statistical analysis). Most of the teachers used bibliographic resources for the preparation of the activities, the most common being PubMed/Medline and Ocenet or Elsevier. None of the individuals used intelligent systems, or at least domain specific ones, but general tools for searching clinical practices, such as YouTube or a Google search. This also led to the conclusion that using specific tools for searching domain concepts is very useful for preparing and executing CBL activities and may also be relevant for brighter students to make it easier for them to find information (active student learning). Figure 5 shows the results for testing the perceptions of the three systems in this case-based learning context (CBL). It shows the classification of the three systems (i.e., first, second, and third place) by the respondents in terms of their perceived usefulness in teaching. The number that appears above the bar plots indicates the number of respondents who ranked each of the systems first (left), second (middle), and third (right).

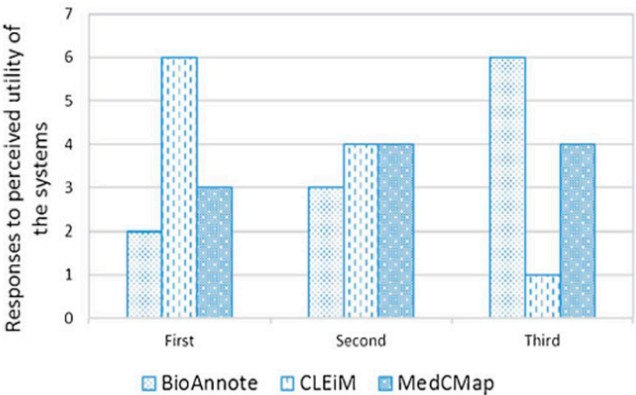

**Figure 5.** Classification of the three systems (i.e., first, second, and third place) by the respondents in terms of their perceived usefulness in teaching. The number that appears above the bar plots indicates the number of respondents who ranked each of the systems first (left), second (middle), and third (right), (single column fitting image).

Despite none of the teachers knowing about the natural language processing capabilities, they found their use interesting for preparing and executing learning activities. Table 3 shows the responses to the perceived utility of the systems, CLEiM being valued well by the users.

**Table 3.** Subjective perceptions of the BioAnnote, CLEiM and MedCMap systems.

| System | Statements | Answer | | | | | Mode | Median |
|--------|-----------|---------------------------|------------|-----------|---------|---------------------|------|--------|
| | | 1 Strongly Disagree | 2 Disagree | 3 Neutral | 4 Agree | 5 Strongly Agree | | |
| BioAnnote | Q1 | 1 | 0 | 1 | 5 | 4 | 4 | 4 |
| | Q2 | 0 | 0 | 1 | 2 | 8 | 5 | 5 |
| Cleim | Q1 | 0 | 0 | 1 | 5 | 5 | 4–5 | 4 |
| | Q2 | 0 | 0 | 0 | 3 | 8 | 5 | 5 |
| MedCMap | Q1 | 0 | 0 | 3 | 2 | 6 | 5 | 5 |
| | Q2 | 0 | 0 | 2 | 2 | 7 | 5 | 5 |

The overall conclusion is that this type of intelligent information access system helps the integration of knowledge into learning activities for both teachers and students. Moreover, due to teachers still not knowing the full capabilities of this type of technology applied to the bio-science teaching area, there is still the need for effort towards bridging the gap between what can be done and how it can be applied to improve learning success.

## 7. Conclusions

The present work proposes and defines an architecture for the integration of information—structured, semi-structured, and unstructured—from heterogeneous sources and provides complementary information access via recommendations. The platform brings together different technologies, including the automatic processing of texts, the recognition of named concepts, sentiment analysis, and the use of intelligent agents. In addition, the architecture of the platform adheres to Semantic Web standards according to W3C criteria. The result is an agile, modular system that is easily adaptable to different domains and offers access to complementary information and knowledge for analyzing complex texts. These characteristics have been tested on searching and learning tasks for tourism and biomedicine domains. The obtained results prove that end-users were able to perform the search tasks with a 2:1 ratio of success versus other baseline systems, and the end user perception on usability and capabilities greatly outperformed other systems. The end user experience regarding the enrichment functionalities was also positively evaluated, indicating an adequate selection of

the recommended complementary information. After all these evaluations, we can also conclude that the platform can be useful to the general population, expert groups, and researchers.

**Author Contributions:** Conceptualization, R.M.G., M.d.B.R., F.A.G., D.G.P. and E.G.-C.; methodology, R.M.G., M.d.B.R., F.A.G., D.G.P. and E.G.-C.; software, R.M.G., M.d.B.R., F.A.G., D.G.P. and E.G.-C.; validation, R.M.G., M.d.B.R., F.A.G., D.G.P. and E.G.-C.; formal analysis, R.M.G., M.d.B.R., F.A.G., D.G.P. and E.G.-C.; investigation, R.M.G., M.d.B.R., F.A.G., D.G.P. and E.G.-C.; resources, R.M.G., M.d.B.R., F.A.G., D.G.P. and E.G.-C.; data curation, R.M.G., M.d.B.R., F.A.G., D.G.P. and E.G.-C.; writing—original draft preparation, R.M.G., M.d.B.R., F.A.G., D.G.P. and E.G.-C.; writing—review and editing, R.M.G., M.d.B.R., F.A.G., D.G.P. and E.G.-C.; visualization, R.M.G., M.d.B.R., F.A.G., D.G.P. and E.G.-C.; supervision, R.M.G., M.d.B.R., F.A.G., D.G.P. and E.G.-C.; project administration, R.M.G., M.d.B.R., F.A.G., D.G.P. and E.G.-C.; funding acquisition,. All authors have read and agreed to the published version of the manuscript.

**Funding:** The author(s) disclosed receipt of the following financial support for the research, authorship, and/or publication of this article: This work has been partially supported by the Spanish Ministry of Science and Innovation through the National Programme for Research Aimed at the Challenges of Society (IPHealth, TIN-2013-47153-C3-1), Universidad Europea de Madrid, Universidad de Alcalá, Universidad Francisco de Vitoria and Universidad Politécnica de Madrid.

**Institutional Review Board Statement:** Not applicable.

**Informed Consent Statement:** The designed and implemented system was evaluated with students, teachers and technical team volunteers (simulating the potential users), and they had full knowledge of all the details of the investigation and agreed to participate in the system evaluation process.

**Data Availability Statement:** Not applicable.

**Acknowledgments:** This publication has emanated from research supported in part by iPHEALTH. Smart Platform based on Open, Linked, and Big Data for decision-making and learning in the health field. Financing entity: Ministerio de Economía y Competitividad. Society Challenges Plan. iPHealth (TIN-2013-47153-C3-1) and MA2VICMR improving the access, analysis, and visibility of information and multilingual and multimedia content online for the Community of Madrid (S2009/TIC-1542). Financing entity: Comunidad de Madrid IV Regional Plan for Scientific Research and Innovation.

**Conflicts of Interest:** The authors declare no conflict of interest.

## Appendix A. Performance Tasks

### Appendix A.1. Task 1

Plan a weekend excursion in Spain or to a nearby European capital using the information you harvest on destinations, attractions, and accommodation. First, do your planning using www.visiteurope.com (assessed on 8 August 2021), and then do the same using the proposed platform. Write a text using the information collected, citing your chosen destination and the places you can visit. Note how long the searches took to find the needed information. You have 10 minutes to use each platform. Then, answer the first nine questions of the questionnaire (see Appendix B), providing answers for both platforms.

### Appendix A.2. Task 2

You will view some tourism videos. Members of Group A will use YouTube: https://www.youtube.com/watch?v=9HcKUlyG5l8 https://goo.gl/73tkU5 (assessed on 8 August 2021) https://www.youtube.com/watch?v=0sYNHhKqIdQ https://goo.gl/oVUdcX (assessed on 8 August 2021) https://www.youtube.com/watch?v=NxZsUcC2AGo https://goo.gl/wCgAar (assessed on 8 August 2021) and members of Group B the proposed platform (locations provided). You will then make a conceptual map for both experiences (to see why this is necessary, see Novak & Gowin [1988]). You have 20 minutes to perform this task. You should: record the time needed to complete the task, assess the information provided in the videos and the number of times you needed to watch them, assess any recommendations made, and assess your motivation to visit

the places suggested/search for further information. Then, answer the remaining six questions of the questionnaire.

**Appendix B. Questionnaire**

Q1. Is the interface friendly?

Q2. Does the system make the information easy to find?

Q3. Is it comfortable to use the interface and platform?

Q4. Is the velocity of the platform reasonable?

Q5. Is it easy to learn its use?

Q6. Does the platform help you to find the information you look for?

Q7. Is the platform faster than a traditional search system?

Q8. Would you use the platform again?

Q9. Are you confident with the results provided by the platform?

Q10. How useful do you find the natural language processing and concept recognition functionalities?

Q11. How useful do you find the information integration, sources, and system information retrieval from different sources (Freebase, DBPedia, Expedia, and Trip Advisor)?

Q12. How useful do you find the depth of the retrieved information in terms of showing detailed information of different concepts?

Q13. How useful do you find the basic recommendation that indicates whether a concept is recommendable or not?

Q14. How useful do you find the advanced suggestions depending on user preferences?

Q15. How useful do you find the concept evaluation that allows end-users to provide feedback to the system?

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
