# Peer review of "A Domain-Adaptable Heterogeneous Information Integration Platform: Tourism and Biomedicine Domains"

_information, doi:10.3390/info12110435_

Round 1

Reviewer 1 Report

The paper presents the architecture of the Intelligent Domain-Adaptable Platform to collect information from different sources. The architecture is validated in tourism and medical domains.

However, the information about Big data possibilities in this domain is missed.

Also, it would be great to add the description of the audience involved in questionary.

The paper presents the architecture of the Intelligent Domain-Adaptable Platform to collect information from different sources. The platform is developed to collect semistructured and structured information. Natural language processing is supported too. The architecture is validated in tourism and medical domains using questionary. The paper is well structured and written in an understandable form. The conclusions are supported by the results and explained the practical importance of the paper.

However, the information about Big data possibilities in this domain is missed. Also, it would be great to add a description of the audience involved in the questionary. I am not sure about the number of responders (11 teachers) for technology evaluation. Is it enought to assess the platform?

In addition, please describe which tools for ontology representation are used in the platform and how do you compare two ontologies.

Author Response

The paper presents the architecture of the Intelligent Domain-Adaptable Platform to collect information from different sources. The architecture is validated in tourism and medical domains.

However, the information about Big data possibilities in this domain is missed.

A new paragraph has been included on line 122 and 3 new references.

[67] García-Santa N., García-Cuesta E., Villazón-Terrazas B. (2015) Controlling and Monitoring Crisis. In: Gandon F., Guéret C., Villata S., Breslin J., Faron-Zucker C., Zimmermann A. (eds) The Semantic Web: ESWC 2015 Satellite Events. ESWC 2015. Lecture Notes in Computer Science, vol 9341. Springer, Cham. https://doi.org/10.1007/978-3-319-25639-9_9

[68] Hassan Zadeh, A., Zolbanin, H.M., Sharda, R. et al. Social Media for Nowcasting Flu Activity: Spatio-Temporal Big Data Analysis. Inf Syst Front 21, 743–760 (2019). https://doi.org/10.1007/s10796-018-9893-0

[69] Kamdar, M.R., Fernández, J.D., Polleres, A. et al. Enabling Web-scale data integration in biomedicine through Linked Open Data. npj Digit. Med. 2, 90 (2019). https://doi.org/10.1038/s41746-019-0162-5

Also, it would be great to add the description of the audience involved in questionary.

An explanatory paragraph has been added on the type of audience involved in the evaluation of the system. (Lin 653)

The paper presents the architecture of the Intelligent Domain-Adaptable Platform to collect information from different sources. The platform is developed to collect semistructured and structured information. Natural language processing is supported too. The architecture is validated in tourism and medical domains using questionary. The paper is well structured and written in an understandable form. The conclusions are supported by the results and explained the practical importance of the paper.

However, the information about Big data possibilities in this domain is missed.

 Also, it would be great to add a description of the audience involved in the questionary. I am not sure about the number of responders (11 teachers) for technology evaluation. Is it enought to assess the platform?

A reference to the evaluation of this system, that was published in the International Journal of Medical Informatics, has been introduced.

[64] Aparicio, F., Morales-Botello, M. L., Rubio, M., Hernando, A., Muñoz, R., López-Fernández, H., ... & de Buenaga, M. (2016). Perceptions of the use of intelligent information access systems in university level active learning activities among teachers of biomedical subjects. International journal of medical informatics, 112,

In addition, please describe which tools for ontology representation are used in the platform and how do you compare two ontologies.

We added the tool that we used in line 541

Reviewer 2 Report

The paper is potentially interesting, considering the attempt of empirical experimentation/validation. I personally believe that the scientific soundness of the paper could be improved as follows: 1) Many key concepts are not properly referred. I would suggest, among others, to add proper references the concept of formal ontology in Computer Science (typically by Guarino) and Semantic Web (normally by Barnes-Lee). 2) The technical foundation of the work relies mostly on NPL and Semantic Web technology. The former is not properly introduced and discussed. The latter could be significantly extended by considering also Linked Data and the relevance of integration issues in social platforms (e.g. Web 2.5 and similar). 3) Recommendation functionalities also seem to play a relevant role in the context of the proposed architecture but a background discussion is missed. 4) The experimental assessment adds value to the paper but the explanation of underlying methodologies and the discussion of results should be improved. 5) Last but not least, I got the feeling that the presentation could be improved to result more fluid to readers.

Author Response

The paper is potentially interesting, considering the attempt of empirical experimentation/validation. I personally believe that the scientific soundness of the paper could be improved as follows: 1) Many key concepts are not properly referred. I would suggest, among others, to add proper references the concept of formal ontology in Computer Science (typically by Guarino) and Semantic Web (normally by Barnes-Lee).

Requested references have been added (lines 58 and 214)

[65] Berners-Lee, T., Hendler, J., & Lassila, O. (2001). The semantic web. Scientific american, 284(5), 28-37.

[66] Guarino, N., Oberle, D., & Staab, S. (2009). What is an ontology?. In Handbook on ontologies (pp. 1-17). Springer, Berlin, Heidelberg.

2) The technical foundation of the work relies mostly on NPL and Semantic Web technology. The former is not properly introduced and discussed. The latter could be significantly extended by considering also Linked Data and the relevance of integration issues in social platforms (e.g. Web 2.5 and similar).

A short introductory paragraph on NLP has been added on line 75 and in line 89 an introductory paragraph on Linked Data.

3) Recommendation functionalities also seem to play a relevant role in the context of the proposed architecture but a background discussion is missed.

The recommendation functionalities and technology used is provided in section 2.3, for clarification purposes we have rewritten the background adding a short paragraph in the introduction and added an introductory sentence in section 2.3.  Recall that the recommendation is part of the domain application presented in the paper and it is a plug and play functionality.

4) The experimental assessment adds value to the paper but the explanation of underlying methodologies and the discussion of results should be improved.

We added explanatory paragraph of results in line 691

5) Last but not least, I got the feeling that the presentation could be improved to result more fluid to readers.

Reviewer 3 Report

In this submission, the authors present a recommendation platform, consisting of a number of components, that could be adapted to multiple domains, exploiting various information sources (structured, semi-structured and unstructured). They also provide two example cases (tourism and bio-medicine domains) and perform an online evaluation of their platform.

Overall, the novelty of the proposed platform is not assessed, especially w.r.t other similar platforms referenced in the related work section. In this respect, the experimental part is not helpful, as it compares domain-specific recommenders instead of comparing platforms. Additionally, the experimental part is problematic due to the very limited and very homogenous number of participants involved (especially in the case of the tourism platform). Lastly, the “baselines” in both domains are not typically recommendation systems, even though they offer filtering capabilities.

Finally, the authors do not publicly disclose the platform, but only the two studied domains (tourism and biomedicine) via a SourceForge link that indicates that both models have been developed at least six years ago.

Author Response

Indeed, the platform began to be developed 6 years ago, but substantial improvements have been introduced in successive implementations and it has not been released because that was not the purpose but rather to experiment with various technologies for the integration of information from different domains. The experimental part is not as extensive as would have been desired, basically due to budgetary limitations, but it gives an idea of the potential of the developed platform. The complete evaluation of this system, in medical domain, was published in the International Journal of Medical Informatics.  

[64] Aparicio, F., Morales-Botello, M. L., Rubio, M., Hernando, A., Muñoz, R., López-Fernández, H., ... & de Buenaga, M. (2016). Perceptions of the use of intelligent information access systems in university level active learning activities among teachers of biomedical subjects. International journal of medical informatics, 112, 21-33.

Round 2

Reviewer 2 Report

I believe there is still room for improvement.

However, the feedback received has been somehow addressed.

Reviewer 3 Report

In this revised version of the manuscript, the authors have not addressed any of the issues raised in my initial review. Therefore, my decision regarding the acceptance of the manuscript has not changed.